# *In Situ* Processing and Efficient Environmental Detection (iSPEED) of tree pests and pathogens using point-of-use real-time PCR

Arnaud Capron[1], Don Stewart[2], Kelly Hrywkiw[1], Kiah Allen[1], Nicolas Feau[1], Guillaume Bilodeau[3], Philippe Tanguay[2], Michel Cusson[2,4], Richard C. Hamelin[1,4,5]*

**1** Department of Forest and Conservation Sciences, The University of British Columbia, Vancouver, British Columbia, Canada, **2** Natural Resources Canada, Canadian Forest Service, Laurentian Forestry Centre, Québec, Canada, **3** Canadian Food Inspection Agency, Ottawa, Ontario, Canada, **4** Institut de Biologie Intégrative et des Systèmes (IBIS), Université Laval, Québec, Canada, **5** Département des sciences du bois et de la forêt, Faculté de Foresterie et Géographie, Université Laval, Québec, Canada

* richard.hamelin@ubc.ca

**Data Availability Statement:** All relevant data are within the paper and its Supporting Information files.

## Abstract

Global trade and climate change are responsible for a surge in foreign invasive species and emerging pests and pathogens across the world. Early detection and surveillance activities are essential to monitor the environment and prevent or mitigate future ecosystem impacts. Molecular diagnostics by DNA testing has become an integral part of this process. However, for environmental applications, there is a need for cost-effective and efficient point-of-use DNA testing to obtain accurate results from remote sites in real-time. This requires the development of simple and fast sample processing and DNA extraction, room-temperature stable reagents and a portable instrument. We developed a point-of-use real-time Polymerase Chain Reaction system using a crude buffer-based DNA extraction protocol and lyophilized, pre-made, reactions for on-site applications. We demonstrate the use of this approach with pathogens and pests covering a broad spectrum of known undesirable forest enemies: the fungi *Sphaerulina musiva*, *Cronartium ribicola* and *Cronartium comandrae*, the oomycete *Phytophthora ramorum* and the insect *Lymantria dispar*. We obtained positive DNA identification from a variety of different tissues, including infected leaves, pathogen spores, or insect legs and antenna. The assays were accurate and yielded no false positive nor negative. The shelf-life of the lyophilized reactions was confirmed after one year at room temperature. Finally, successful tests conducted with portable thermocyclers and disposable instruments demonstrate the suitability of the method, named *in Situ* Processing and Efficient Environmental Detection (iSPEED), for field testing. This kit fits in a backpack and can be carried to remote locations for accurate and rapid detection of pests and pathogens.

## Introduction

The increase in DNA sequences in public databases, driven by advances in DNA sequencing technologies, has revolutionized the molecular diagnostics of pests and pathogens of crops and

**Funding:** RCH was funded by information Genome Canada Large Scale Applied Research Program #10106 and by a Genome Canada Genomics Application Partnership Program # 6102.

**Competing interests:** The authors have no competing interests exist.

trees. Molecular identification and detection have become essential components of the prevention, mitigation and management toolbox of forest pests and pathogens [1,2]. DNA testing allows the rapid, sensitive and accurate detection of target organisms from small amounts of environmental samples harvested during surveys or inspections. This can entirely bypass the time-consuming culture or rearing steps that were previously required to perform a valid identification. The standardization of DNA barcoding databases for fungi and insects [3,4] has generated extensive DNA sequence data that can be exploited to design taxon-specific DNA assays [5–7]. New approaches make use of whole genomes to identify diagnostic genome regions that are translated into highly accurate detection assays [8,9].

The Polymerase Chain Reaction (PCR) is a powerful method to amplify DNA fragments. The development of instruments that can measure fluorescence in real-time [10] allows the design of single-step DNA detection assays, removing the need to visualize the PCR product by gel electrophoresis based on fragment size. The use of fluorescent DNA binding dyes such as SYBR green [11] or the use of internal probes labelled with a dye [12] enables more complex assay creation. For example, multiplexing can be achieved by using different dyes on probes targeting different genes or organisms. There are now thousands of PCR assays for the detection of pests and pathogens [13,14], and several are used operationally [15].

One drawback of PCR-based detection until recently is that it needs to be conducted in a laboratory environment with benchtop and sensitive instruments (thermal cyclers, centrifuges, pipettes) to handle the small volumes of reagents. Most chemicals and reagents used in the reactions are typically not stable at room temperature for long periods and must be maintained at 4°C or frozen. DNA extraction methods generally require multiple steps and non-portable equipment. They usually comprise a mechanical disruption step, often in a homogenizer, followed by lysis of the sample. DNA is then purified from the lysate using liquid-liquid extraction or bound to a silica gel matrix before washing and elution. These processes are time-consuming and costly and require multiple buffers for lysis, washes and elution as well as several instruments: water bath, centrifuge and several pipettes for different volumes. Similarly, real-time PCR reagents typically require storage at -20°C. Furthermore, the reaction setup involves pipetting microliter-scale volumes multiple times.

Having the capacity to perform PCR on-site would be beneficial for environmental applications for tests that have to be performed in the field. This can be an advantage for the detection and identification of forest invasive pests and pathogens because sampling is often conducted in remote locations. Sending samples to a laboratory and obtaining results often requires several days. Field-based DNA testing would speed up the process and could inform additional sampling strategies, for example if the presence of a threatening pest or pathogen is discovered. This could trigger more rapid mitigation and management actions, crucial aspects of prevention [16]. However, the complexity of the extraction and amplification processes and the cost of sample processing have hampered the adoption of such testing.

In recent years, medical point-of-care and military point-of-need applications of DNA tests have generated interest in designing small real-time PCR instruments, ranging from transportable, such as the Coyote Bioscience Mini8 Plus [17] and the Bio Molecular Systems Mic [18], to handheld such as the Biomeme two3 and Franklin [19]. Herein we describe the development of a complete sample-to-data, point-of-use, real-time PCR system to provide *in situ* Processing and Efficient Environmental Detection (iSPEED) of pests and pathogens. This system can easily be carried in a backpack and is ideal for environmental applications. It comprises a minimal DNA extraction method to process environmental samples without centrifuge or homogenizer instruments and room-temperature stable ready-to-use real-time PCR reagents that we used in portable instruments. As a show-case, we demonstrate successful detection and identification of a variety of forest pests and pathogens: the fungi *Sphaerulina musiva*,

*Cronartium comandrae* and *Cronartium ribicola*, the oomycete *Phytopththora ramorum* and the insects *Lymantria dispar dispar* and *Lymantria dispar asiatica*.

## Material & methods

### Plant, fungal and insect material

All material was obtained as part of surveys of tree diseases or insect pests or via artificial inoculation of plant material. The number of extractions performed or tests conducted are reported in the extraction or results sections. We tested direct PCR detection of pathogens on leaves and stems of naturally-infected poplars and pines. Poplar leaves infected with *Sphaerulina musiva*, causal agent of the Septoria poplar leaf spot and canker, were collected from a plantation in Salmon Arm, British Columbia in 2015 as described previously [20] (Fig 1A). A leaf disk (about 4 mg) was cut out from a poplar hybrid (*P. trichocarpa* x *P. deltoides*) leaf infected with the pathogen *S. musiva* and further cut in two halves (Fig 1A); each half was used for comparison of two DNA extraction methods (described below).

Aeciospores of the Comandra blister rust fungus of pines, *Cronartium comandrae*, were collected in May 2018 from stems and branches of lodgepole pines (*Pinus contorta*) in Smithers, British Columbia, and stored at 4°C, 31% relative humidity, for 7 days before being transferred to -20°C. Aeciospores of the white pine blister rust fungus, *Cronartium ribicola*, were collected in April 2019 from Cypress Mountain, British Columbia and held at 4°C until used (Fig 1F). Approximately 2.5 mg of spores was used per extraction.

For *Phytophthora ramorum*, the causal agent of sudden oak and sudden larch death, we inoculated leaves of *Rhododendron macrophyllum*, *R. purdomii* and *Rhododendron spp.* collected at the University of British Columbia Botanical Garden (Fig 1B). Using a scalpel, a 0.5–1.0 cm surface incision was made on the abaxial surface of surface-sterilized detached leaves, taking care not to puncture the adaxial side of the leaf. Then, an agar plug of a *P. ramorum* culture [NA2 lineage (04–38813)] [21] grown on carrot agar at 23°C in the dark for 10 days [22] was placed over the wound mycelium side face down. Inoculated leaves were placed on damp wipes in a metal tray and enclosed in a plastic bag with a breathable patch to maintain air circulation and humidity. The leaves were incubated in the dark for 6 days under high humidity at 19–21°C, after which a 5 mm leaf disc (about 20 mg) was cut from the edge of the necrotic zone. We also used DNA extracted from cultured mycelia of NA1 PR-11-010, described in Feau *et al.* [23], as a positive control.

Pinned insects belonging to the genus *Lymantria* were obtained from collectors from Lorgues, France, in 2007, and from Krutinka in the Omsk Oblast, Russia, in 2000 [7] (Fig 1C and 1D). In addition, samples were obtained from *Lymantria* pheromone traps deployed in Vancouver, BC, in 2016 as part of the annual gypsy moth survey by the Canadian Food Inspection Agency (CFIA, Fig 1E). Either a single leg or a pair of antennae was used for extraction (1–2.5 mg of material).

### DNA extraction

DNA was extracted using established protocols as well as using a field-ready protocol. DNA was extracted from *S. musiva* lesions cut out from infected poplar leaves and from *Cronartium* spores using column-based DNeasy Plant Mini Kit (Qiagen, Venlo, Netherlands) following the manufacturer's instructions, but modified as follows: disruption was performed by grinding in a MM300 Mixer Mill (Retsch, Haan, Germany) for 2 minutes at 30 Hz, in the presence of three 3 mm steel beads. For *P. ramorum* cultures, total genomic DNA was extracted using the CTAB (cetyl trimethylammonium bromide) method. DNA was eluted in TE (Tris–EDTA) buffer (10 mM Tris-HCL, 0.1 mM EDTA, pH 8) and used as template

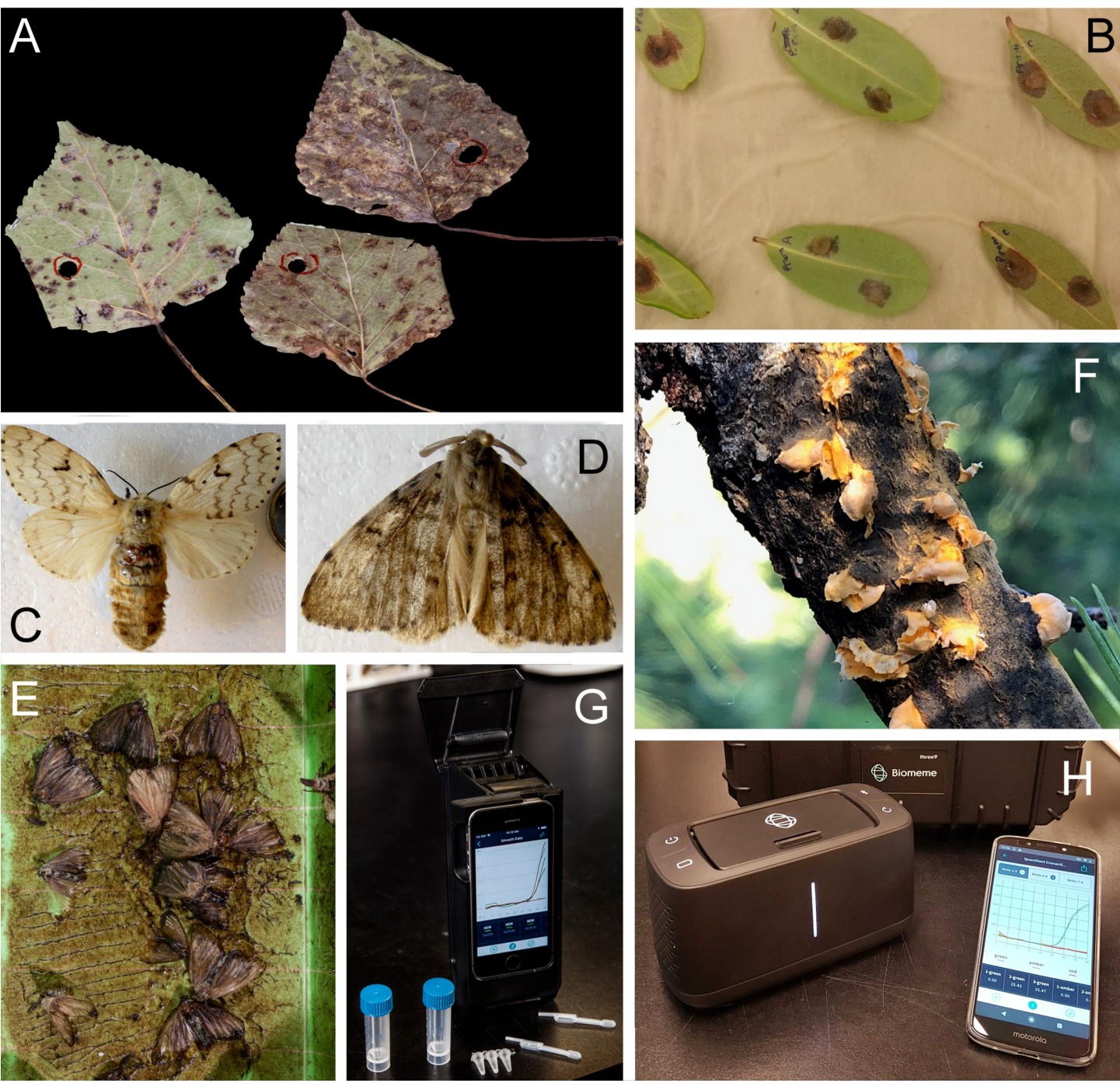

**Fig 1.** Sample-to-data point-of-use real-time PCR system to provide *in situ* processing and efficient environmental detection (iSPEED) of pests and pathogens: A) *Sphaerulina musiva* infected poplar leaves; B) *Phytophthora ramorum* infected rhododendron leaves; C) *Lymantria dispar dispar;* D) *Lymantria dispar asiatica*; E) An example of gypsy moth specimens collected in a pheromone trap; F) *Cronartium ribicola* cankers; G) The two3 (Biomeme), a first-generation hand-held real-time PCR instrument, with the plastic consumables used for field-trials: 5 mL screw cap vials, a PCR strip and 20 μL pastettes; H) The Franklin (Biomeme), a portable real-time PCR instrument used for the Point-Of-Use field assays, along with a smartphone used to parameterize the experiments, monitor results and upload them to a remote server.

[24]. For *Lymantria* samples, DNA was extracted from single or multiple antennae or legs using the DNeasy Blood & Tissue kit (Qiagen, Venlo, Netherlands) following the manufacturer's insect protocol. Disruption was performed by grinding in a MM300 Mixer

Mill (Retsch, Haan, Germany) for 2 minutes at 30 Hz in the presence of a single 5 mm steel bead.

A simplified field-ready cost-effective DNA extraction method was developed using Edwards buffer [25]: Tris 200 mM pH 8.0, EDTA 25 mM, NaCl 250 mM, SDS 0.5% (w/v). For the naturally infected poplar and the inoculated rhododendron leaves, 1% PVPP (w/v) (Millipore Sigma, Darmstadt, Germany) was added to the buffer to mitigate the effect of phenolic compounds present in leaves on PCR reactions. Intact infected plant or insect material was incubated in 40 μL of buffer for 10 min at 95°C to release the nucleic acid into the buffer solution. Any dry block can be used to perform the incubation, including the thermocycler itself. The extract was then used directly as template for PCR, either as a 50-fold or 100-fold dilution in water, depending on the tissue. A Standard Operating Procedure (SOP) for point-of-use real-time PCR is provided as supplementary material.

## Real-Time PCR TaqMan assays

**Real-time PCR assays.** We used real-time PCR assays that have been previously developed, tested and validated for *Sphaerulina musiva*, *P. ramorum* and European gypsy moth (EGM) *Lymantria* (S1 Table). The *S. musiva* assay used the Genome-Enhanced Detection and Identification (GEDI [9]) approach; it amplifies a genome fragment (SepMu) found in *S. musiva*, but absent in its close relatives, plus a fragment of the RuBisCo large subunit gene from the plant DNA (*RbcL*), as an internal control [20]. For detection of *P. ramorum*, we used two assays: (i) TAIGA-C62 (also developed using the GEDI approach), which targets a single copy nuclear region and provides high levels of specificity [23], and (ii) *TrnM*, which amplifies a portion of the mitochondrial genome, providing higher sensitivity [26]. For the gypsy moth, we used a duplex real-time PCR assay that can distinguish between the Asian and European alleles of the FS1 marker [7].

For the rust fungi, we designed an assay that can differentiate the native *C. comandrae* blister rust from the invasive, exotic *C. ribicola* pine rusts and their hybrids [27]. The assay targets the glutamine synthetase gene (*Dcon10*) from *C. ribicola* and *C. comandrae* (NCBI PopSet 63080856). The primers and probes were designed using Primer3 [28,29] and IDT OligoAnalyzer Tool (IDT, Corralville, IA, USA). An Infinity Plus DNA probe using Locked Nucleic Acid (LNA) bases (to increase specificity) was designed for each of *C. ribicola* and *C. comandrae* (S1 Table). The specificity of the assays was tested against DNA samples extracted from *C. ribicola* and *C. comandrae* (S9 Table).

**Laboratory real-time PCR testing using fresh reagents.** Reactions performed in the lab were conducted with the QuantiTect Multiplex PCR master mix (Qiagen, Venlo, Netherlands) in 20 μL, following the manufacturer's instructions. The instrument used was an Applied Biosystems Viia7 thermocycler (ThermoFisher Scientific, Waltham, MA, USA). Cycling parameters were the following: 15 min at 95°C, 40 cycles of 15 sec at 95°C and 90 sec at 60°C, as specified by the manufacturer. Details of the reaction set-up for each assay can be found in S2 Table.

**Field-ready master mixes.** Field-ready reaction mixes used lyophilized reagents that comprised the QuantiTect mix. Primers and probes were added to 10 μL of the mix from the 100 μM stocks, and trehalose was added to a final concentration of 5% from a 30% stock to increase stability of the reagents. The reactions were aliquoted in 100 μL MicroAmp Fast 8-tube strips (ThermoFisher Scientific, Whaltam, MA, USA), frozen at -20°C and then lyophilized for 60 to 90 minutes in a Freezone 2.5 Liter freeze-drier (Labconco, Kansas City, MO, USA). After lyophilization, the strips were stored in the dark, either at room temperature or -20°C. To conduct the real-time PCR assays, the lyophilized reagents were resuspended in a

final volume of 20 μL. For reactions performed with DNA extracted using kits, two microliters of DNA were diluted in 18 μL of water. The reactions prepared using the direct PCR method were resuspended in 20 μL of a 1/50 or 1/100 dilution of the crude DNA extract. The stability of the lyophilized reagents was tested every three months by running real-time PCR with the tubes that were stored frozen and those stored at ambient temperature. Details about each assay are provided in S3 Table. The numbers of replicates are listed in S4–S7 Tables.

**Shelf-life tests.**   Lyophilized strips for *RbcL* and *SepMu* were prepared as described above and stored in the dark at room temperature for the duration of the experiments. The assays were performed using a 1:50 dilution of a crude extract prepared from an infected poplar leaf disk, as described above. The DNA was stored at -20˚C for the duration of the experiment. The tests were performed using a Mini8 Plus thermocycler (Coyote Bioscience, Beijing, PRC), with the following conditions: 95˚C for 15 minutes and 40 cycles of 95˚C for 15 seconds and 60˚C for 1 minute 30. Tests were performed in duplicate.

**Field-ready extraction and real-time PCR using portable instruments.**   Field-ready real-time PCR testing was conducted on the Franklin portable real-time PCR thermocycler (Biomeme, Philadelphia, PA, USA). The DNA extraction was performed on approximately 2 mg of rust aeciospores per reaction. Both the extractions and real-time PCR reactions were performed with 20 μL disposable pipettes (pastettes; Alpha Laboratories, Eastleigh, UK) that were used for dilution of the extractions and pipetting of the samples into real-time PCR strips. Two extractions were performed using the Franklin as a dry block, and the real-time PCR reactions were conducted in duplicate with lyophilized reagents. DNA extracted using a column was used as a positive control. Both extractions were tested in duplicate. The $C_t$ values and the number of technical replicates are listed in S8 Table.

**Data analysis.**   For experiments performed on the Viia7, $C_t$ values were obtained using the QuantStudio Software v1.3 (ThermoFisher Scientific, Waltham, MA, USA). The baseline was set to cycles 3 to 15, and the thresholds were determined manually, making them as low as possible while keeping them above background noise. For experiments performed with the Mini8 Plus, $C_t$ values were determined using the Mini8 Plus software v2.0.14s (Coyote Bioscience, Beijing, PRC). Baselines and thresholds were determined by the software using automatic settings. For reactions performed on the Franklin, $C_t$ values were generated by the Biomeme Cloud (Biomeme, Philadelphia, PA, USA) platform, used in the reprocessing mode to adjust the thresholds as described above. Baselines were determined automatically by the software. For statistical analysis, ANOVAs were performed using R [30], while charts were generated using the R package ggplot2 [31].

# Results

## Assembling a real-time PCR field kit: DNA extraction and real-time PCR reagents

We compared field-ready and column-based extractions as well as fresh and lyophilized reagents in order to assess the performance of the PCR field kit.

**Efficacy of field-ready DNA extraction method on *Sphaerulina musiva* poplar pathogen.**   As a first proof of concept experiment, we compared the efficacy of our field-ready DNA extraction method against a column-based DNA extraction. We chose an assay that can detect *Sphaerulina musiva*, a fungal pathogen that causes leaf spot disease (Fig 1A) and perennial stem and branch cankers in poplars [32].

Both the field-ready and the column-based extractions yielded positive real-time PCR results from infected leaf spots for the internal leaf control and the pathogen assay. Although the cycle threshold ($C_t$) values were lower for the column-based extraction DNA (ANOVA:

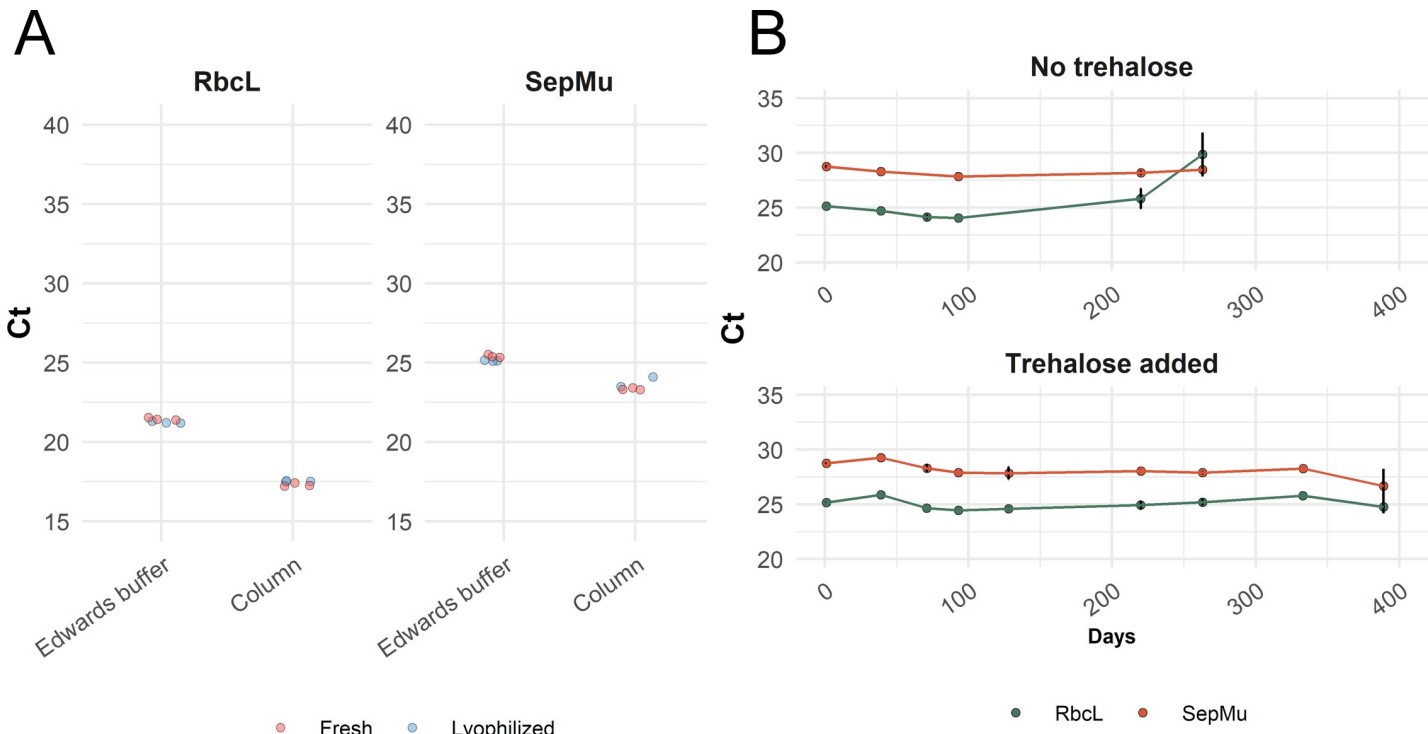

**Fig 2. Real-time PCR detection of *Sphaerulina musiva* from naturally-infected poplar leaves using field-ready and laboratory protocols.** A) $C_t$ values obtained using the plant control *RbcL* (left) and *S. musiva* assays (right), comparing the fresh vs. lyophilized PCR reagents and the two extraction methods; B) shelf-life test using lyophilized reactions with or without trehalose. The probes carried the FAM fluorophore.

F = 119.58, p< 0.001), the field-ready extraction using the Edwards buffer extraction yielded unambiguously positive detection of the pathogen, using either fresh or lyophilized reagents, with $C_t$ values ranging from 21.24 to 25.41 (Fig 2A, S4 Table). These $C_t$ values are comparable to those observed in another study using extraction kits [20]. In comparing field-ready extractions with column-extracted DNA, the higher $C_t$ values observed for the former were expected given the thorough disruption that precedes column-based extractions and the likely presence of inhibitors in the crude extract.

In comparing freshly prepared and freeze-dried reactions, lyophilization did not have a significant impact on $C_t$ values (ANOVA: F = 0.05, p>0.9), indicating that the lyophilized reactions perform adequately.

**Detection of *Phytophthora ramorum* on infected rhododendron leaves.** The second organism tested was *Phytophthora ramorum*, an exotic oomycete pathogen. We assessed DNA detection by real-time PCR for the *P. ramorum*-infected leaf discs (Fig 1B) using DNA extracted with the field-ready Edwards buffer (1:100 dilution), direct real-time PCR, and DNA extracted with a column from a pure culture as baseline. For the field-ready extraction, we placed the leaf disk in Edwards buffer and heated it at 95°C for 5 min before conducting real-time PCR with fresh reagents; the same extraction was used to perform real-time PCR with the lyophilized reagents. DNA from the pure culture and all hosts inoculated with *P. ramorum* produced a detectable amplification signal, with $C_t$ values ranging from 15.77 to 32.16 for the *TrnM* mitochondrial assay and from 21.93 to 34.62 for the TAIGA-C62 nuclear assay; none of the uninoculated leaves yielded detectable signals (Fig 3, S5 Table). All fresh and lyophilized reactions yielded detectable signals, although fresh reactions yielded slightly lower $C_t$ values than lyophilized reactions (F = 28.43, p<0.0001). Cycle threshold values for the *TrnM*

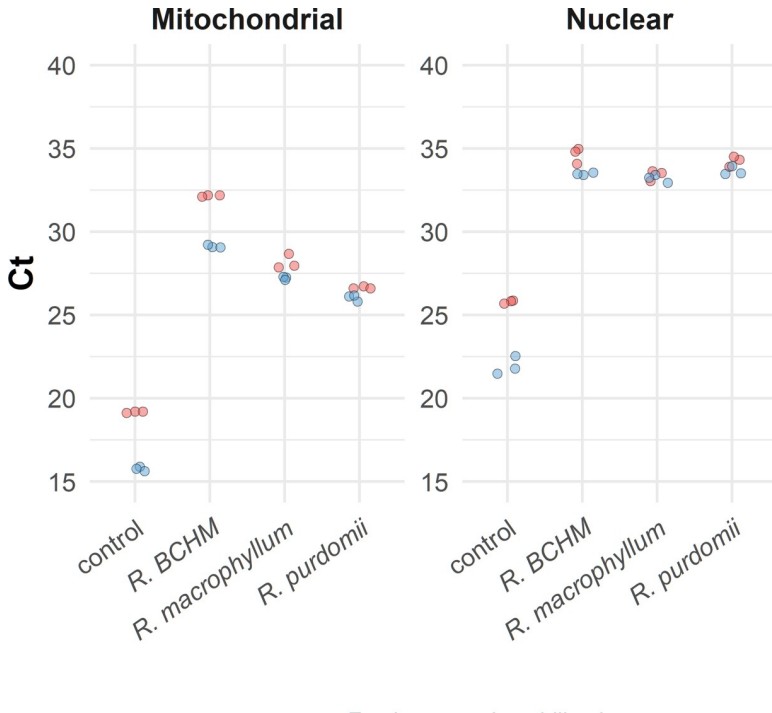

**Fig 3. Real-time PCR amplification of *Phytophthora ramorum* from artificially infected *rhododendron* leaves using field-ready and laboratory protocols.** Cycle threshold values obtained with rhododendron leaves from different species infected with *P. ramorum* as well as DNA extracted from a cultured sample (control). The figure contrasts results obtained with fresh and lyophilized real-time PCR reagents. The *TrnM* mitochondrial assay (left) targets a *P. ramorum* mitochondrial DNA sequence, while the TAIGA-C62 nuclear assay (right) targets a single copy *P. ramorum* nuclear locus. The probes carried the FAM fluorophore.

mitochondrial assay were lower than for the TAIGA-C62 nuclear assay (F = 316.18, p<0.0001). This result is expected because of the higher sensitivity of the multi-copy mitochondrial assay than that of the single-copy nuclear assay [23,26].

**Identification of Asian and European gypsy moths.** To demonstrate the approach using insect material, we selected *Lymantria dispar*, a species complex that includes the European Gyspy Moth (EGM), *Lymantria dispar dispar*, and the invasive Asian Gypsy Moth (AGM), *Lymantria dispar asiatica*. The AGM4 duplex real-time PCR assay targets two alleles of the nuclear marker FS1, each one being the major allele in their respective population [7]. We assessed the performance of our field-ready protocol using this multiplex assay and tested it using pinned and trapped moth specimens. DNA was extracted with the field-ready extraction as well as the column-based kit from a single leg of pinned specimens (Fig 1C and 1D). The EGM sample was positive for the FS1 North American allele using lyophilized and fresh real-time PCR reagents, but was negative for the Asian allele (Fig 4, S6 Table). In contrast, the AGM sample was positive for the Asian FS1 allele and negative for the European allele using both column and field-ready crude DNA samples, and fresh as well as lyophilized real-time reagents (Fig 4, S6 Table). An ANOVA showed no significant difference between fresh and lyophilized reactions or between the extraction methods (F = 3.770, p>0.05).

We tested the field-ready extraction with insects obtained from pheromone traps (Fig 1E, S6 Table). Typically, those traps remain in the field for several weeks and contain tens to hundreds of insects that, in North America, are presumed to be EGM. DNA was extracted from a pair of *Lymantria dispar* antennae using the Edwards buffer and the field-ready protocol as

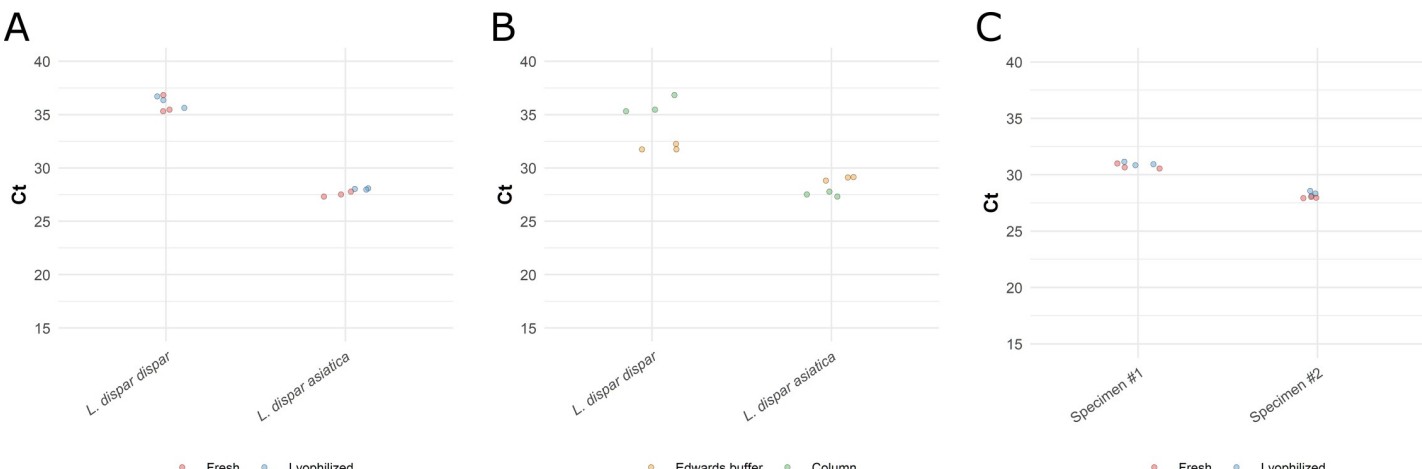

**Fig 4. Real-time PCR detection of *Lymantria dispar* using field-ready and laboratory protocols.** DNA was extracted from *Lymantria dispar* legs or antennae using a Qiagen DNA extraction column and a field-ready protocol using Edwards buffer. The $C_t$ values on panel A and B are for the European FS1 allele for *L. dispar dispar* and the Asian FS1 allele for *L. dispar asiatica*. A) Results of the tests using column extracted DNA with fresh and lyophilized reactions. B) Results of the test comparing column extracted vs. field-ready DNA extraction reporting $C_t$ values for the same individuals. C) Results of the test using samples collected from a pheromone trap, with fresh and lyophilized reagents, the $C_t$ values shown are for the European FS1 allele. The FS1 Asian allele carries the FAM fluorophore and the FS1 North American allele carries the CY5 fluorophore.

described above. Real-time PCR was conducted using both fresh and lyophilized reagents. Both specimens tested amplified only the North American FS1 allele (Fig 4C, S6 Table). An ANOVA showed a small but significant difference of $C_t$ values obtained with fresh vs. lyophilized reagents (F = 9.236, p<0.05).

*Cronartium* **blister rust pathogens of pine.** We tested our protocol on fungal spores obtained in the field from pine rust cankers caused by *Cronartium ribicola* and *Cronartium comandrae*, two closely related rust fungi attacking different hosts (Fig 1F). The assays targeting *C. ribicola* and *C. comandrae* detected these pathogens in all the processed samples, with $C_t$ values ranging from 26.5 to 32.9 (Fig 5A, S7 Table). The different spore samples and reagent conditions did not yield significantly different $C_t$ values for the *C. comandrae* samples, but the DNA extraction methods were significantly different (F = 61.1248, p<0.001); similarly, fresh and lyophilized reagents did not impact $C_t$ values (F = 0.333, p>0.3) in *C. ribicola*, but DNA extraction methods and extraction batch significantly affected $C_t$ values in *C. ribicola* (F = 273.166, p<0.0001 for method and F = 125.214, p<0.0001 for extraction number); this is possibly a reflection of the variation in spore abundance in each blister. Although the thorough mechanical disruption and purification provided by the column-based DNA extraction improves the performance of the DNA for real-time PCR, we found no false negatives, pointing to the high sensitivity of the approach.

**Shelf-life test of the field-ready lyophilized kits.** As the lyophilized mastermix needs to be prepared and aliquoted in advance for use in the field, a crucial aspect of the usefulness of this approach is the long-term stability of the kit without refrigeration. We prepared two batches of PCR strips containing all the PCR reagents for the *S. musiva* assays, with and without trehalose and tested them at regular intervals. The strips were stored at room temperature in the dark and tested over the course of a year, running each assay in triplicate. We did not find a significant difference between the reactions with and without trehalose (F = 1.457, p>0.05) (Fig 2B). However, the reactions conducted without trehalose yielded no amplification of the targets after 263 days at room temperature. By comparison, amplification products were detected for all reactions with added trehalose until the end of the test, on day 389. The

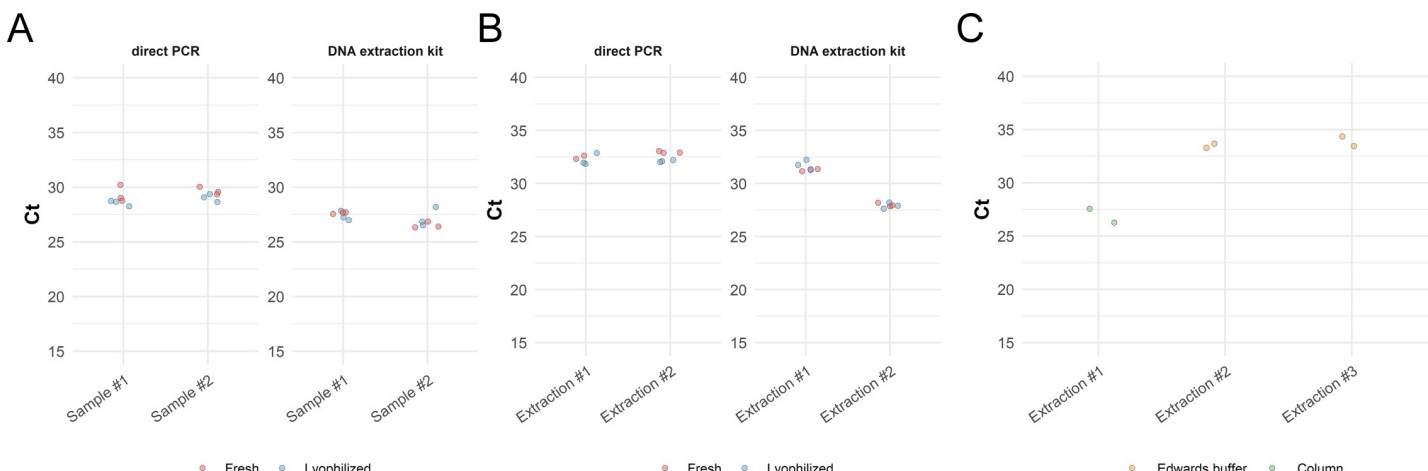

**Fig 5. Real-time PCR amplification of *Cronartium spp*. spores collected on pines.** Average $C_t$ values obtained with *Cronartium* spores, contrasting fresh and lyophilized reagents, and field-ready vs. kit DNA extraction. A) Results using spores from *C. comandrae*. Spores from two different blisters were used, labelled as sample #1 and sample #2. DNA from spores from each sample was extracted using a DNA extraction kit or using the field-ready Edwards buffer. B) Results using spores from *C. ribicola*. Spores were pooled from multiple blisters. Four extractions were performed and tested, two using a DNA extraction kit and two using field-ready Edwards buffer. C) Results obtained using a portable real-time PCR instrument. Two DNA extractions were prepared in the instrument and tested using lyophilized reagents (extraction #2 and #3). DNA extracted using a kit was used as a positive control (extraction #1). The *C. ribicola* probe carries the FAM fluorophore and the *C. comandrae* carries the CY5 fluorophore.

trehalose-supplemented reactions had a small but significant age effect (F = 4.356, p<0.05), but no effect on the consistency of the replicates (F = 4.334, P>0.05). However, there were no false negatives in the reactions with trehalose at any time point.

## Real-time PCR field kit demonstration

To demonstrate the workability of our protocols in field applications, we combined both the field-ready lyophilized reagents and DNA extraction on a portable instrument without the use of additional lab equipment (see Standard Operating Procedure for point-of-use real-time PCR in the Supplementary Material). We used fixed volume, disposable, 20 μL pastettes, pre-aliquoted reagents in disposable containers and the lyophilised *Cronartium* reagents to perform real-time PCR in a portable Franklin real-time PCR unit (Biomeme), an instrument capable of performing 9 concurrent reactions, monitored on 3 channels, such as for FAM, CY5 and Texas Red, allowing the multiplexing of up to three assays (Fig 1H). Spores of *C. ribicola* were prepared as described earlier. Both DNA samples extracted with the field ready protocol using the Edwards buffer yielded positive amplifications, with $C_t$ values ranging between 32 and 33.5 (Fig 3C, S8 Table). Testing was also conducted on a two3 instrument (Fig 1G) (Biomeme) using the *S. musiva* and the Lymantria assays with similar results.

## Discussion

Forestry and environmental applications often require sampling in remote areas far away from laboratory facilities. The need for on-site DNA testing capability is growing in areas such as forest health monitoring, where detection of potentially invasive species can require immediate actions. In this proof-of-concept study, we demonstrated the rapid development of real-time PCR assays for point-of-use pest and pathogen detection. Although assays conducted in the laboratory with fresh reagents on thoroughly homogenized samples extracted with the kit were more sensitive, the iSPEED assays did not yield false negatives or false positives. Field-testing using the protocols described here provides a useful and efficient on-site screening that can be

complementary to more extensive follow-up lab testing. All the material required to perform the assays and the instrument can be carried in a small backpack, demonstrating the potential of performing simple, rapid, cost-effective, user-friendly and accurate molecular testing in the field at the point-of-use. This proof-of-concept demonstrates the feasibility of implementing on-site testing that can provide rapid detection of forest pests and pathogens. Once a species of interest is identified in the field, additional sampling and testing can be performed for confirmation with additional assays.

In addition to the availability of a portable real-time PCR instrument, the creation of iSPEED required that we develop a crude, quick and simple DNA extraction method that can be used in the field. For environmental applications, DNA extractions should also be inexpensive. Simple methods have been developed previously for blood [33], fixed tissue samples [34], insects [35] and fungi [36] among others. Our field DNA extraction method follows a simple protocol that does not require instruments (beyond the real-time thermocycler) and uses a basic, inexpensive Tris-based buffer (Edwards buffer) containing SDS [25]. This method was chosen due to its effectiveness, simplicity and very low-cost. The other challenge in bringing real-time PCR assays to the field is the stability of reagents at ambient temperature so that refrigeration is not required. Our initial trials with lyophilized real-time PCR master mixes containing the polymerase, reaction buffer, dNTP, primers and probe showed that the reagents could tolerate the process, allowing us to pre-combine all required PCR reagents, thus reducing operations to the addition of the sample extracts diluted in water. Strip PCR tubes, pre-prepared with trehalose added to reach 5% of the reaction volume, ensured stability since trehalose is a cryoprotectant [37] as well as a PCR enhancer [38].

To demonstrate the utility of the method, we chose to test it on a range of organisms, including fungi, oomycetes and insects. The target organisms were selected as examples because they cover a significant part of the tree of life and represent threats to forests and urban trees. The fungus *Sphaerulina musiva* is a significant threat to hybrid poplars grown in plantations in North America [39,40]. It has recently established in new regions, including western North America, where it threatens poplar plantations [20], but it is still absent from Europe where regulatory agencies monitor its presence. Fungi of the *Cronartium* genus are important pathogens that cause cankers and blisters on the stems and branches of pines. *Cronartium ribicola* is responsible for the white pine blister rust disease that contributed to the decline of North American populations of five-needle pines and the listing of whitebark pine as an endangered species [41]. *Cronartium comandrae* can attack two- and three-needle pines such as *Pinus banksiana*, *P. contorta* and *P. ponderosa* and can also cause mortality in these pines that are widely distributed in Canada and are of commercial importance for lumber and pulp production [42]. The oomycete *P. ramorum* is a pathogen under quarantine or regulated in multiple countries, including Canada [43], the USA [44] and the European Union [45]. *Phytophthora ramorum* causes large, deadly, epidemics on oaks in the USA [46] and larch in the UK [47] and is capable of infecting a wide range of hosts, including many ornamental plants, such as rhododendron [48].

We also demonstrated the use of field-ready DNA on subspecies of a lepidopteran insect, the gypsy moth. Gypsy moth caterpillars are defoliators that feed on multiple tree species, especially oak (*Quercus* spp.), trembling aspen (*Populus tremuloides*) and birch (*Betula* spp.) [49]. The European gypsy moth (EGM), *Lymantria dispar dispar*, is established in eastern North America [50]. The Asian gypsy moth (AGM), *Lymantria dispar asiatica*, represents a higher risk to North America since it has a broader host range than EGM and their females can fly up to 25 km during the mating season. It is present in several countries in Asia, but has not yet become established in North America. Efforts to prevent its introduction and establishment involve surveys and inspections which, in turn, require reliable identification. One concern is

that AGM and EGM can hybridize and could produce flight-capable hybrids that may escape identification [51]. Morphological identification of adults and larvae of the two subspecies is difficult at best and impossible for eggs.

Other portable assays have been developed for plant pathogens; however we think that the one reported here presents multiple advantages relative to earlier methods: 1) the ability to leverage the vast library of existing published and validated real-time PCR assays, 2) the extensive ecosystem supporting real-time PCR: multiple enzymes and chemistry, numerous types of assays and a growing range of portable instruments, 3) flexibility of design, and 4) cost-effectiveness. Loop-mediated isothermal amplification (LAMP) assays are popular for field applications and allow rapid and sensitive detection of plant pathogens [52] and insects [53]. However, LAMP reagents and instruments have more limited availability and assay design remains complex and difficult. While a real-time PCR assay only needs a pair of primers and an optional internal probe, LAMP assays require at least four specific primers in close proximity, with six providing better results. Finally, while multiplexing real-time PCR assays is relatively straightforward with the use of different fluorophores carried by the probes, LAMP multiplexing has relied on differences in melting temperatures of products [54], chemical modifications [55] or innovative oligonucleotide use [56], among others. Here we demonstrated the broad applicability of the iSPEED protocols: they can be used on different types of material, ranging from infected deciduous tree leaves, evergreen perennial rhododendron leaves, fungal rust spores and legs and antennae of moths. Further testing will be required for more recalcitrant materials. For example, those containing wood with extractives will be more challenging and might require adjustments to the current protocols.

Although we developed our protocols with forest pest and pathogen applications in mind, they could also be applied to detect the presence of other invasive species, such as the American bullfrog (*Lithobates catesbeianus*) or the red-eared slider (*Trachemys scripta elegans*) by quickly detecting their DNA from water samples. They could also be used to confirm the species of fish sold at fish markets, retail stores and restaurants, as it is frequently mislabeled [57]. Finally, there are also potential medical applications, for example, the detection of food-borne pathogens like *Clostridium perfringens*, for which a large number of real-time PCR assays have been developed [58–63].

Future work should focus on improving and expanding the scope of available assays. Preliminary testing indicates real-time PCR master mixes from manufacturers other than those of the Qiagen company used here are also suitable. The DNA extraction step could also be improved to increase sensitivity or expand the range of material suitable for testing. For example, the use of Chelex 100 resin or simple disruption tools such as micro-pestles and/or abrasives could be tested to improve yield. Also, more capable portable real-time PCR instruments are in development, with increased block capacity and additional optical channels. Those new instruments will allow the development of more complex assays through multiplexing as well as providing extra wells, increasing throughput or offering redundant testing for added accuracy. Our work highlights the great potential of *in situ* point-of-use real-time PCR in improving our ability for early detection and management of forest enemies.

## Supporting information

**S1 Table. Oligonucleotide sequences used in qPCR assays.** The table lists all the oligonucleotides used in the various tests. Probes were ordered with a FAM fluorophore for simplex reactions and with the FAM and CY5 combination for duplexes. The + sign denotes a LNA base. (DOCX)

**S2 Table. Reagents used in fresh reactions.** The table lists the reagents used to prepare the fresh reactions for each assay.
(DOCX)

**S3 Table. Reagents used in lyophilized reactions.** The table lists the reagents used to prepare the lyophilized strips for each assay.
(DOCX)

**S4 Table. Real-time PCR amplification of *Sphaerulina musiva* directly from naturally-infected poplar hybrid leaves.** DNA was extracted from leaves of poplar hybrids (*P. tricho-carpa* x *P. deltoides)* infected by *S. musiva* using a Qiagen DNA extraction column and a field-ready protocol using Edwards buffer. DNA amplification was conducted in triplicate by qPCR using field-ready lyophilized and fresh reagents. Average $C_t$ values of the replicates are reported for each of the conditions tested with the plant internal control (RbcL) and the *S. musiva* (SepMu) assays. All tests were conducted using material from the same leaf disc to allow direct comparisons between extraction methods. Both probes used carry the FAM fluorophore.
(DOCX)

**S5 Table. Real-time PCR amplification of *Phytophthora ramorum* from artificially infected rhododendron leaves.** DNA was extracted from rhododendron leaves using a Qiagen DNA extraction column and a field-ready protocol using Edwards buffer. A pure culture of *P. ramorum* was also extracted with a Qiagen extraction protocol as a positive control. DNA amplification was conducted in triplicate by qPCR using field-ready lyophilized reagents and fresh reagents. Average $C_t$ values of the replicates are reported for each of the conditions tested with mitochondrial and nuclear *P. ramorum* assays. All tests were conducted using material from the same leaf disc. Both probes used carry the FAM fluorophore. NA = No Amplification.
(DOCX)

**S6 Table. Real-time PCR amplification of *Lymantria dispar* from adult legs and antennae.** DNA was extracted from *Lymantria dispar* legs or antennae using a Qiagen DNA extraction column and a field-ready protocol using Edwards buffer. DNA amplification was conducted in triplicate by qPCR using field-ready lyophilized reagents and fresh reagents. Average $C_t$ values and standard deviations are reported for each of the conditions tested with a multiplex assay targeting the Asian and North American allele at the FS1 locus. The FS1 Asian allele carries the FAM fluorophore and the FS1 North American allele carries the CY5 fluorophore. NA = No Amplification.
(DOCX)

**S7 Table. Real-time PCR results for *Cronartium* spp spores.** DNA was extracted from *Cronartium comandra* and *C. ribicola* spores using a Qiagen DNA extraction column and a field-ready protocol using Edwards buffer. DNA amplification was conducted in triplicate by qPCR using field-ready lyophilized reagents and fresh reagents. Average $C_t$ values and standard deviations are reported for each condition tested with the *Cronartium* assays. The *C. ribicola* probe carries the FAM fluorophore and the *C. comandrae* carries the CY5 fluorophore.
(DOCX)

**S8 Table. Real-time PCR results using the portable real-time PCR instrument.** Average $C_t$ values obtained using the Cronartium assays on DNA extracted from *C. ribicola* aeciospores in the Franklin instrument. The *C. ribicola* probe carries the FAM fluorophore.
(DOCX)

**S9 Table. Specificity tests for the *Cronartium* assays.** Average $C_t$ values obtained using the Cronartium assays with *C. ribicola* and *C. comandrae* DNA. One hybrid was tested as well. A) results obtained with the assays used as simplex; B) results obtained with the assays used as a duplex. The *C. ribicola* probe carries the FAM fluorophore and the *C. comandrae* probe carries the HEX fluorophore. NA = No Amplification.
(DOCX)

**S1 Data. Standard operating procedure for point-of-use real-time PCR.**
(DOCX)

# Acknowledgments

The authors would like to thank Xianya "Sabrina" Qu for testing the protocol and Karen McLachlan Hamilton for providing the pheromone traps. We would like to thank Justyna Nowakowska for reviewing our manuscript.

# Author Contributions

**Conceptualization:** Arnaud Capron, Richard C. Hamelin.

**Investigation:** Arnaud Capron, Kelly Hrywkiw.

**Resources:** Don Stewart, Kiah Allen, Nicolas Feau, Guillaume Bilodeau, Philippe Tanguay, Michel Cusson.

**Validation:** Arnaud Capron.

**Visualization:** Arnaud Capron.

**Writing – original draft:** Arnaud Capron, Richard C. Hamelin.

**Writing – review & editing:** Arnaud Capron, Don Stewart, Nicolas Feau, Guillaume Bilodeau, Philippe Tanguay, Michel Cusson.

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
