## [Decision Letter · Decision Letter 0]

2 Jan 2020

PONE-D-19-33958

In Situ Processing and Efficient Environmental Detection (iSPEED) of pests and pathogens of trees using point-of-use real-time pcr

PLOS ONE

Dear Dr Hamelin,

Thank you for submitting your manuscript to PLOS ONE. After careful consideration, we feel that it has merit but does not fully meet PLOS ONE’s publication criteria as it currently stands. Therefore, we invite you to submit a revised version of the manuscript that addresses the points raised during the review process.

Broadly, I think many readers will recognize the utility of the iSPEED platform. Rapid, DNA-based field detection of insects and pathogens is sorely needed by the forest health community. However, it is important to get the details right. The manuscript is not yet suitable for publication, but can be improved sufficiently by addressing the reviewers’ comments. In particular, you must address the following points:

1) Both reviewers noted that some of the citations are formatted improperly, and as a result, some works are missing from the references list. Please correct.

2) I agree with Reviewer 1 that a fair amount of material in the Results section should be moved to the Introduction or Discussion.

3) I also agree with Reviewer 1 that additional details are required about the simplified DNA extraction methods, qPCR assay setup, and any cross-reactivity tests that were performed.

4) There are some typographical and grammatical errors scattered throughout the manuscript (e.g., “…AGM as a broader host range than EGM…”, 1st paragraph, p. 12; “…significant effect age effect…”, 1st paragraph, p.15). Reviewer 1 has provided a marked-up copy that identifies other minor errors. Please check the manuscript carefully and revise as necessary.

We would appreciate receiving your revised manuscript by Feb 16 2020 11:59PM. To enhance the reproducibility of your results, we recommend that if applicable you deposit your laboratory protocols in protocols.io, where a protocol can be assigned its own identifier (DOI) such that it can be cited independently in the future. For instructions see: http://journals.plos.org/plosone/s/submission-guidelines#loc-laboratory-protocols

We look forward to receiving your revised manuscript.

Kind regards,

Frank H. Koch, PhD

Academic Editor

PLOS ONE

Journal Requirements:

1. 

This work was funded by Genome Canada, Genome British Columbia, Genome Quebec, the

Canadian Forest Service and the Canadian Food Inspection Agency, through a Genomics

Applications Partnership Program (GAPP 6102; Genome Canada) grant. Also this work is funded

by Genome Canada, Genome British Columbia, Genome Quebec, the Canadian Forest Service,

the Canadian Food Inspection Agency and FP Innovations through a Large Scale Applied Research

Project (LSARP 10106; Genome Canada) grant.

RCH was funded by information Genome Canada Large Scale Applied Research Program #10106 and by a Genome Canada Genomics Application Partnership Program # 6102.

Reviewers' comments:

Reviewer's Responses to Questions

**Comments to the Author**

1. Is the manuscript technically sound, and do the data support the conclusions?

Reviewer #1: Yes

Reviewer #2: Yes

2. Has the statistical analysis been performed appropriately and rigorously? 

Reviewer #1: Yes

Reviewer #2: Yes

3. Have the authors made all data underlying the findings in their manuscript fully available?

Reviewer #1: Yes

Reviewer #2: Yes

4. Is the manuscript presented in an intelligible fashion and written in standard English?

Reviewer #1: Yes

Reviewer #2: Yes

5. Review Comments to the Author

Reviewer #1: The authors present a simplified DNA extraction procedure and a mobile real-time PCR platform for detecting DNA of selected fungal pathogens and an invasive insect for in-field testing. The authors present novel qPCR assays for detection of the rust pathogens Cronartium ribicola and C. comandrae, and also adapted published assays for Phytophthora ramorum, Septoria musiva, and Lymantria spp. to the mobile qPCR platform. A marked up copy of the manuscript is attached.

Reviewer's Specific Comments:

1. The Results section contains a substantial amount of background information/discussion about the various diseases/pests that would be more appropriate for the Introduction and/or Discussion sections.

2. The description of the simplified DNA extraction method and the field-ready qPCR assay setup are both missing crucial details, (e.g., the amount of tissue that was put into the extraction, the pH of the extraction buffer, the model of the mobile thermocycler [found elsewhere in manuscript], the number of biological/technical replicates for the qPCR reactions, how the threshold fluorescence value was determined, statistical analysis details).

3. It is unclear if any testing on cross-reactivity of the Cronartium spp. assays was tested. Given that the polymorphism targeted for the specific probes is just 2 of 14 nucleotides within the probe, and the primers are common, it is likely that the two assays may cross react on the non-target template DNA of the opposite species. As one purpose for the test stated in the manuscript is to identify hybrids between the two species, the specificity of the assays must be tested and the data presented.

4. The discussion of the iSPEED platform vs. isothermal detection methods (i.e. LAMP) should be more thoroughly developed and expanded. The reviewer acknowledges the authors' points regarding the applicability of existing qPCR assays with the mobile qPCR platform, but the authors do not specifically discuss the challenges/limitations of LAMP assay development.

5. The authors need to define the abbreviation "NA" used in the tables in the supplemental data. The reviewer assumes this designation indicates the target was not detected. The authors also need to specify which qPCR probes were labeled with FAM and which were labeled with CY5.

Reviewer #2: The article is original, has good technical quality and large general interest. Many forest ecosystems suffer from the presence of pests and pathogens, very often unpredictable in changing climate conditions. International and national phytosanitary regulations also stipulate measures aimed at managing the risk of introducing harmful organisms, often following pest-risk analysis based on scientific evidence. The large number of plants traded and the inconspicuousness of many pests, in particular microbial pathogens, highlight the need to mitigate the risk associated also with the intercontinental trade pathways (Eschen et al. 2018). Real-time PCR assays based on the TaqMan system have been recently developed for the identification of many pests and pathogens e.g. Phytophthora sp., including P. cactorum, P. megasperma, P. plurivora, P. pseudosyringae and P. quercina that cause significant damage to forest ecosystems (Nowakowska et al. 2016). The development of molecular methods has become powerful tool which facilitates diagnosis, i.e. PCR methods aiding species identification, making the process more fast and accurate. The paper describes very useful method of rapid and reliable environmental detection of tree pests and pathogens thanks to the real-time PCR. The prevention measures can then be undertaken in order to mitigate the development of the harmful organisms in forest stands.

About the content:

The title, abstract and keywords clearly reflect paper's content.

Introduction presents the problem clearly.

Experimental methods are adequate. The markers were appropriately chosen for the species detection and the sensitivity of each primer + probe set was tested. The idea of lyophilization of the PCR reaction mixtures ensures the stability of the reagents in the field conditions. The applied appropriate concentration of trehalose into the PCR mixtures ensured the long term stability (more than 260 days) of the reagent kit at room temperature.

Discussion and results are justified.

References are adequate but incomplete. Please check the bibliography citation in the text (on p. 4, 5, 8, 9, 11 and 17) because some of them are missing in the literature at the end of the MS. All citations should have numbers in the MS.

About Presentation:

Length is commensurate with the paper's content.

Quality of tables is adequate but quality of figures 4 & 5 could be improved.

Some data could be improved:

- Check the spelling of the Latin names, e.g. “Clostridium perfrigens” should be “Clostridium perfringens” on p. 17, or “C. comandrae” in Supplementary Table S7

- “thedetection” on p. 17,

About Scientific evaluation:

The general scientific approach is properly stated and well explained. A good quality tool (iSPEED protocol) has been developed and its usefulness for the chosen insects and pathogen detection in situ has been statistically proven. Of course, the field tests will not replace laboratory tests, but instead, provide an effective screening which may speed up prevention measures undertaken in a forest stand. Thanks to such techniques, it is also possible to monitor the biodiversity of the investigated species. The increasing number of new alien pests and pathogens of woody plants in many parts of the world will certainly promote the development of fast, reliable and costless techniques of the harmful organism detection in the field.

Therefore, I recommend publishing the article after minor corrections.

6. PLOS authors have the option to publish the peer review history of their article (what does this mean?). If published, this will include your full peer review and any attached files.

Reviewer #1: No

Reviewer #2: Yes: Justyna Nowakowska

---

## [Author Response · Author response to Decision Letter 0]

21 Feb 2020

Editor’s comments: 

1) Both reviewers noted that some of the citations are formatted improperly, and as a result, some works are missing from the references list. Please correct.

=>The citations were properly formatted and we verified the content of the reference list. 

2) I agree with Reviewer 1 that a fair amount of material in the Results section should be moved to the Introduction or Discussion.

=>We moved most of the pathogen and pest biology material to the Discussion. We did keep only the essential information in the Results to improve understanding. 

3) I also agree with Reviewer 1 that additional details are required about the simplified DNA extraction methods, qPCR assay setup, and any cross-reactivity tests that were performed.

=>We added detailed information about the simplified DNA extraction in the M&M. In addition, two new sections were added: shelf-life test and data analysis. We also present a new table with the samples used for cross-reactivity. 

4) There are some typographical and grammatical errors scattered throughout the manuscript (e.g., “…AGM as a broader host range than EGM…”, 1st paragraph, p. 12; “…significant effect age effect…”, 1st paragraph, p.15). Reviewer 1 has provided a marked-up copy that identifies other minor errors. Please check the manuscript carefully and revise as necessary.

=>We have proof-read the latest version and we think that we fixed all of the typos. 

Reviewer’s comments

Reviewer #1: The authors present a simplified DNA extraction procedure and a mobile real-time PCR platform for detecting DNA of selected fungal pathogens and an invasive insect for in-field testing. The authors present novel qPCR assays for detection of the rust pathogens Cronartium ribicola and C. comandrae, and also adapted published assays for Phytophthora ramorum, Septoria musiva, and Lymantria spp. to the mobile qPCR platform. A marked up copy of the manuscript is attached.

Reviewer's Specific Comments:

1. The Results section contains a substantial amount of background information/discussion about the various diseases/pests that would be more appropriate for the Introduction and/or Discussion sections.

=>We moved most of the pathogen and pest biology material to the Discussion. We did keep only the essential information in the Results to improve understanding. 

2. The description of the simplified DNA extraction method and the field-ready qPCR assay setup are both missing crucial details, (e.g., the amount of tissue that was put into the extraction, the pH of the extraction buffer, the model of the mobile thermocycler [found elsewhere in manuscript], the number of biological/technical replicates for the qPCR reactions, how the threshold fluorescence value was determined, statistical analysis details).

=>We agree that some details were missing from the methods. We added all necessary details in the current version in the M&M. In addition, we added two new sections to clarify the testing: shelf-life test and data analysis. Furthermore, a complete SOP is provided in the supplementary material.

3. It is unclear if any testing on cross-reactivity of the Cronartium spp. assays was tested. Given that the polymorphism targeted for the specific probes is just 2 of 14 nucleotides within the probe, and the primers are common, it is likely that the two assays may cross react on the non-target template DNA of the opposite species. As one purpose for the test stated in the manuscript is to identify hybrids between the two species, the specificity of the assays must be tested and the data presented.

=>The reviewer is raising a valid point. To increase specificity and ensure the differentiation of the two closely related fungi, we designed probes modified with Locked Nucleic Acid (LNA). The LNA bases used in probes greatly enhance their specificity (Single nucleotide polymorphism genotyping using locked nucleic acid (LNA), 2003. Mouritzen P, Nielsen AT, Pfundheller HM, Choleva Y, Kongsbak L, Møller S. Expert Rev Mol Diagn. Jan;3(1):27-38. DOI: 10.1586/14737159.3.1.27) and have been widely used. We did perform cross-reactivity tests, which did not reveal any cross-amplification in the samples available (see Table S9). The rust assays have also been used to perform hundreds of tests using samples of both species collected across BC (Allen K. Evaluating the presence and introgression of the hybrid forest pathogen Cronartium x flexil. The University of British Columbia. 2019. Available from https://dx.doi.org/10.14288/1.0380539). No cross-reactivity was observed in this extensive survey. 

4. The discussion of the iSPEED platform vs. isothermal detection methods (i.e. LAMP) should be more thoroughly developed and expanded. The reviewer acknowledges the authors' points regarding the applicability of existing qPCR assays with the mobile qPCR platform, but the authors do not specifically discuss the challenges/limitations of LAMP assay development.

=>This is a valid comment. We have now expanded the discussion on the limitation of LAMP: ‘LAMP assays are popular for field applications and allow rapid and sensitive detection of plant pathogens [50] and insects [51]. However, LAMP reagents and instruments have more limited availability and assay design remains complex and difficult. While a real-time PCR assay only requires a pair of oligonucleotide primers and an optional internal probe, LAMP assays require at least four specific primers in close proximity, with six providing better results. Finally, while multiplexing real-time PCR assays is relatively straightforward with the use of different fluorophores carried by the probes, LAMP multiplexing has relied on melting temperature difference of products [52], chemical modifications [53] or innovative oligonucleotide use [54], among others.’ 

5. The authors need to define the abbreviation "NA" used in the tables in the supplemental data. The reviewer assumes this designation indicates the target was not detected. The authors also need to specify which qPCR probes were labeled with FAM and which were labeled with CY5.

=>Fluorophores added in the legends, NA abbreviation explained in table legend: NA = No Amplification.

Reviewer #2: The article is original, has good technical quality and large general interest. Many forest ecosystems suffer from the presence of pests and pathogens, very often unpredictable in changing climate conditions. International and national phytosanitary regulations also stipulate measures aimed at managing the risk of introducing harmful organisms, often following pest-risk analysis based on scientific evidence. The large number of plants traded and the inconspicuousness of many pests, in particular microbial pathogens, highlight the need to mitigate the risk associated also with the intercontinental trade pathways (Eschen et al. 2018). Real-time PCR assays based on the TaqMan system have been recently developed for the identification of many pests and pathogens e.g. Phytophthora sp., including P. cactorum, P. megasperma, P. plurivora, P. pseudosyringae and P. quercina that cause significant damage to forest ecosystems (Nowakowska et al. 2016). The development of molecular methods has become powerful tool which facilitates diagnosis, i.e. PCR methods aiding species identification, making the process more fast and accurate. The paper describes very useful method of rapid and reliable environmental detection of tree pests and pathogens thanks to the real-time PCR. The prevention measures can then be undertaken in order to mitigate the development of the harmful organisms in forest stands.

About the content:

The title, abstract and keywords clearly reflect paper's content.

Introduction presents the problem clearly.

Experimental methods are adequate. The markers were appropriately chosen for the species detection and the sensitivity of each primer + probe set was tested. The idea of lyophilization of the PCR reaction mixtures ensures the stability of the reagents in the field conditions. The applied appropriate concentration of trehalose into the PCR mixtures ensured the long term stability (more than 260 days) of the reagent kit at room temperature.

Discussion and results are justified.

References are adequate but incomplete. Please check the bibliography citation in the text (on p. 4, 5, 8, 9, 11 and 17) because some of them are missing in the literature at the end of the MS. All citations should have numbers in the MS.

=>Good point, all references were verified and reformatted.

About Presentation:

Length is commensurate with the paper's content.

Quality of tables is adequate but quality of figures 4 & 5 could be improved.

=>Unfortunately, this problem has to do with the PLoS One PDF generator that reduces figure quality. The figures we submitted are of high quality and this should not be an issue. 

Some data could be improved:

- Check the spelling of the Latin names, e.g. “Clostridium perfrigens” should be “Clostridium perfringens” on p. 17, or “C. comandrae” in Supplementary Table S7

- “thedetection” on p. 17,

=>Good point, we fixed all of the typos that we could find.

About Scientific evaluation:

The general scientific approach is properly stated and well explained. A good quality tool (iSPEED protocol) has been developed and its usefulness for the chosen insects and pathogen detection in situ has been statistically proven. Of course, the field tests will not replace laboratory tests, but instead, provide an effective screening which may speed up prevention measures undertaken in a forest stand. Thanks to such techniques, it is also possible to monitor the biodiversity of the investigated species. The increasing number of new alien pests and pathogens of woody plants in many parts of the world will certainly promote the development of fast, reliable and costless techniques of the harmful organism detection in the field.

Therefore, I recommend publishing the article after minor corrections.

---

## [Editor Report · Decision Letter 1]

2 Mar 2020

PONE-D-19-33958R1

In Situ Processing and Efficient Environmental Detection (iSPEED) of pests and pathogens of trees using point-of-use real-time pcr

PLOS ONE

Dear Dr Hamelin,

Thank you for submitting your manuscript to PLOS ONE. After careful consideration, we feel that it has merit but does not fully meet PLOS ONE’s publication criteria as it currently stands. Therefore, we invite you to submit a revised version of the manuscript that addresses the points raised during the review process.

**From the Academic Editor: I appreciate the many changes you made in response to my comments and those of the reviewers. At this point, the manuscript is nearly suitable for publication. I went through the revised version (and the supporting information) carefully, and identified a small number of further edits I would like you to make. They are all minor, and rather than list them here, I've attached track-change versions of both the main text and supporting information documents as guidance.

I have one specific request: Please go through the reference list and ensure that all capitalization, italics and formatting are correct. If at all possible, use standard journal name abbreviations. As you probably know, PLOS ONE doesn't perform further editing once a manuscript is accepted. **

We would appreciate receiving your revised manuscript by Apr 16 2020 11:59PM. To enhance the reproducibility of your results, we recommend that if applicable you deposit your laboratory protocols in protocols.io, where a protocol can be assigned its own identifier (DOI) such that it can be cited independently in the future. For instructions see: http://journals.plos.org/plosone/s/submission-guidelines#loc-laboratory-protocols

A marked-up copy of your manuscript that highlights changes made to the original version. This file should be uploaded as separate file and labeled 'Revised Manuscript with Track Changes'.An unmarked version of your revised paper without tracked changes. This file should be uploaded as separate file and labeled 'Manuscript'.A rebuttal letter will not be necessary.

We look forward to receiving your revised manuscript.

Kind regards,

Frank H. Koch, PhD

Academic Editor

PLOS ONE

---

## [Author Response · Author response to Decision Letter 1]

13 Mar 2020

We have done the final small edits requested, including revising the reference formats.

---

## [Editor Report · Decision Letter 2]

17 Mar 2020

*In Situ* Processing and Efficient Environmental Detection (iSPEED) of tree pests and pathogens using point-of-use real-time PCR

PONE-D-19-33958R2

Dear Dr. Hamelin,

We are pleased to inform you that your manuscript has been judged scientifically suitable for publication and will be formally accepted for publication once it complies with all outstanding technical requirements.

With kind regards,

Frank H. Koch, PhD

Academic Editor

PLOS ONE

Additional Editor Comments (optional):

Thank you for completing one more round of edits. I believe the manuscript is now suitable for publication. However, if you're given the chance to revisit the manuscript, I noticed a few very minor things you should consider fixing (they're cosmetic, but if I was an author I'd want to know about them).

P. 14, first paragraph -- "...yielded a positive real-time PCR results..." (singular-plural disagreement, either delete "a" or change "results" to "result")

P. 20, last line -- "homogeneized" should be "homogenized"; insert "the" before "kit"

References 43 and 44 -- "Phytophthora ramorum" should be italicized

S6 Table caption -- insert "using" after "antennae"